# RETHINKING OUT-OF-DISTRIBUTION DETECTION IN VISION FOUNDATION MODELS

## ABSTRACT

Pre-trained vision foundation models have transformed many computer vision tasks. Despite their strong ability to learn discriminative and generalizable features–crucial for out-of-distribution (OOD) detection, their impact on this task remains underexplored. Motivated by this gap, our study investigates vision foundation models in OOD detection. Our findings show that even without complex designs, a pre-trained DINOv2 model, utilizing a simple scoring metric and no fine-tuning, outperforms all prior state-of-the-art models, which typically depend on fine-tuning with in-distribution (ID) data. Furthermore, while the pre-trained CLIP model struggles with fine-grained OOD samples, DINOv2 excels, revealing the limitations of CLIP in this setting. Building on these insights, we explore how foundation models can be further optimized for both ID classification and OOD detection when ID data is available for fine-tuning. From a model perspective, we propose a Mixture of Feature Experts (MoFE) module, which partitions features into subspaces. This mitigates the challenge of tuning complex data distributions with limited ID data and enhances decision boundary learning for classification. From a data perspective, we introduce a Dynamic-$\beta$ Mixup strategy, which samples interpolation weights from a dynamic beta distribution. This adapts to varying levels of learning difficulty across categories, improving feature learning for more challenging categories. Extensive experiments and ablation studies demonstrate the effectiveness of our approach, significantly outperforming baseline methods.

## 1 INTRODUCTION

The task of out-of-distribution (OOD) detection (Sehwag et al., 2021; Liang et al., 2018; Hsu et al., 2020; Lee et al., 2018b) aims to equip models with the capability to discern whether input images originate from unknown OOD classes or belong to in-domain (ID) classes. Mainstream OOD detection methods (Du et al., 2023; Tao et al., 2023; Du et al., 2022; Lee et al., 2018a) focus on learning features and classifiers (Sun & Li, 2022; Liang et al., 2018; Sun et al., 2021; Wang et al., 2022a) from ID data and then develop a score metric (Hendrycks et al., 2019; Sun et al., 2022; Liu et al., 2020; Hendrycks & Gimpel, 2017) to determine whether a sample belongs to ID or OOD. Despite significant advancements, the fundamental challenge in OOD detection is establishing a feature space with high discriminative capacity that can effectively distinguish OOD samples from ID samples, which is still unresolved. Recently, vision foundation models (Maxime et al., 2023; Radford et al., 2021; Kirillov et al., 2023; Singh et al., 2023) trained on large-scale datasets have demonstrated the ability to learn robust and generalizable features, benefiting numerous tasks (Yang et al., 2024; Zhang et al., 2022; Li et al., 2022; Tian et al., 2021). This raises the question: with such powerful models and feature representations, does OOD detection remain a problem?

Although some studies have explored the use of CLIP for OOD detection, a comprehensive comparative analysis of vision foundation models remains lacking. To address this, we begin by investigating whether vision foundation models can effectively serve as OOD detectors and their limitations. In this study, we focus on two representative models: CLIP, trained on 400 million text-image pairs using a text-image contrastive loss, and DINOv2, trained on 142 million web-curated images using self-supervised learning. We conduct an analysis using ImageNet-1K as ID data and OOD data from four datasets (see Tab. 1). Without any model tuning, we directly utilize the features from the models for OOD evaluation. Our results reveal that with a simple KNN metric, DINOv2 surpasses leading OOD detectors that have been fine-tuned on ID data across all evaluated OOD datasets (see

Tab. 1). Additionally, our study highlights that CLIP's pre-trained feature space is less effective for fine-grained tasks like iNaturalist, where DINOv2 performs significantly better (see Tab. 1 and Fig. 1). We hope these findings will inspire further research in this area.

While vision foundation models have achieved impressive performance in OOD detection, there is still room for improvement, particularly on in-domain data with large semantic spaces (e.g., 29.27% FPR95 on the ImageNet-1K OOD benchmark). This prompts us to investigate whether foundation models can be further optimized, leveraging available ID data to improve both ID classification and OOD detection. However, as the number of semantic classes increases, the complexity of the decision boundaries required to distinguish between ID and OOD data grows as well. This heightened complexity creates challenges when fine-tuning foundation models on limited ID data, often forcing a trade-off between model fitting on ID data and preserving generalizable features. Our empirical experiments support this observation: when fine-tuning the DINOv2 pre-trained model on ImageNet-1K ID data, its performance declined on three out of four OOD datasets (see Tab. 1). This suggests that as the model adapts to the ID data distribution, fine-tuning on a complex in-domain distribution with limited ID data can compromise the discriminative and generalizable features crucial for effective OOD detection.

To address these challenges, we propose a Mixture-of-Feature-Expert (MoFE) module, which divides the complex semantic space into multiple subspaces, with each expert specializing in a specific subspace. This approach reduces the difficulty of fitting complex data distributions from limited data by breaking the problem into smaller subproblems. This division eases the optimization process while maintaining the generalizability of features (see Fig. 4 in Appx. A). MoFE operates by partitioning the original feature space into $K$ subspaces based on feature similarities within the ID dataset. Each subspace is assigned to a dedicated expert, and a router assigns samples to the appropriate expert based on these partitions. During training, the router is supervised by the partition assignments to ensure accurate sample-to-expert mapping. Importantly, in our design, each expert focuses solely on optimizing features within its designated partition. This helps prevent interference between features from different partitions, preserving feature diversity and generalizability. Additionally, given that data augmentation has been shown to enhance generalization during fine-tuning, we introduce a novel Mixup data augmentation strategy to further improve feature learning for ID classification and OOD detection with vision foundation models. Our design is based on the observation that different categories exhibit varying levels of learning difficulty. In the raw feature space of vision foundation models, some categories shows great discriminativeness (see Fig. 1d and Fig. 1e), while others do not (see Fig. 1f). For categories that are already well-represented, synthesizing dissimilar samples via vanilla Mixup can blur the decision boundary between ID and OOD, leading to degraded performance (see Fig. 3 in Appx. A). Thus, unlike existing Mixup strategies that treat all categories equally, our approach makes Mixup weight sampling category-dependent by adjusting the sampling distribution (*i.e.* beta distribution) dynamically, taking into account their learning difficulties.

Extensive experiments on existing benchmark datasets demonstrate consistent performance improvements. Notably, we achieve a significant performance boost of +11.33% in FPR95 on the challenging ImageNet-1K dataset, highlighting the effectiveness of our approach in enhancing vision foundation models. Our **contributions** can be summarized as follows:

- We are the first to conduct a comprehensive study on pre-trained foundation models for OOD detection. The promising results emphasize that discriminative and generalizable features are the most important factors for effective OOD detection. Notably, DINOv2 and CLIP exhibit distinct behaviors when handling fine-grained datasets, underscoring the potential advantages of the learning paradigm of DINOv2 .

- To tailor pre-trained vision foundation models for improved ID classification and OOD detection during fine-tuning, we introduce a novel MoFE module to effectively fit ID data distribution and preserve generalizable features, along with a Dynamic-$\beta$ Mixup strategy to enhance generalization and boost OOD detection performance.

- We establish new baselines for OOD detection using foundation models. Our extensive experimental results demonstrate the effectiveness of the proposed model, achieving significant improvements over several competitive baseline methods.

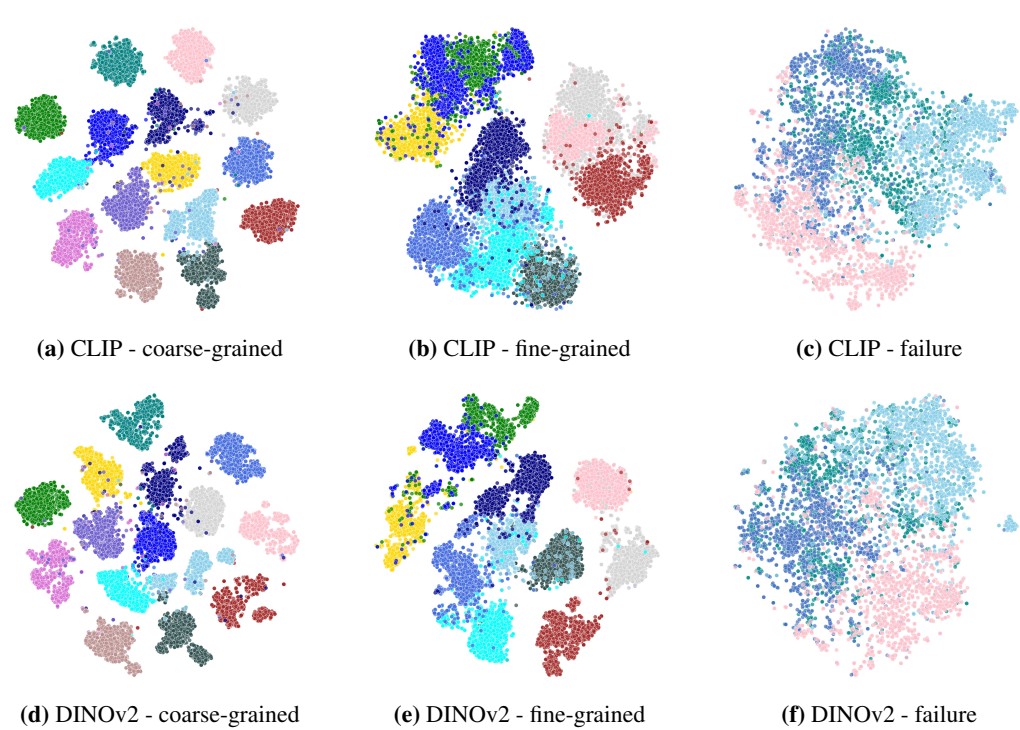

**(a)** CLIP - coarse-grained  **(b)** CLIP - fine-grained  **(c)** CLIP - failure

**(d)** DINOv2 - coarse-grained  **(e)** DINOv2 - fine-grained  **(f)** DINOv2 - failure

**Figure 1: Feature Space Visualization for Foundation Models.** The first row shows the feature space for CLIP and the second is for DINOv2. For each of them, we visualize the features of coarse-grained categories, fine-grained categories, and some failure cases. For the coarse-grained feature visualization (column 1), we randomly select 15 categories from different super classes in ImageNet-1k following WordNet. For the fine-grained feature visualization (column 2), we randomly select 11 fine-grain categories under 3 different super classes. For the failure case visualization, we select the categories which have the low in-domain accuracy.

## 2 PILOT STUDY

In this section, we first introduce preliminaries for the OOD detection task in Sec. 2.1. Then, we explore the impact of foundation models on OOD detection performance and analyze their strengths and weaknesses in Sec. 2.2.

### 2.1 PRELIMINARIES

We consider supervised multi-class classification, where $\mathcal{X}$ represents the input image space and $\mathcal{Y} = \{1, 2, ..., C\}$ represents the label space. The training dataset $\mathbb{D}_{in} = \{(\mathbf{x}_i, y_i)\}_{i=1}^{n}$ is drawn independently and identically distributed (*i.i.d.*) from the joint data distribution $P_{\mathcal{XY}}$. Let $\mathcal{P}_{in}$ denote the marginal distribution on $\mathcal{X}$. Let $f : \mathcal{X} \mapsto \mathbb{R}^{|\mathcal{Y}|}$ be a neural network trained on samples drawn from $P_{\mathcal{XY}}$ to output a logit vector, which is used to predict the label of the input sample.

**Out-of-distribution Detection.** When deploying a machine learning model in real-world scenarios, it is crucial for a reliable classifier not only to accurately classify known in-distribution (ID) samples, but also to recognize any out-of-distribution (OOD) inputs as "unknown". This can be accomplished by incorporating an OOD detector alongside the classification model $f$. OOD can be formulated as a binary classification task. During testing, the objective is to determine whether a sample $\mathbf{x} \in \mathcal{X}$ belongs to $\mathcal{P}_{in}$ (ID) or not (OOD). This decision can be made using a scoring metric $S(\mathbf{x})$:

$$G_\lambda(x) = \begin{cases} \text{ID} & S(\mathbf{x}) \geq \lambda \\ \text{OOD} & S(\mathbf{x}) < \lambda \end{cases}, \tag{1}$$

where samples with higher scores $S(\mathbf{x})$ are classified as ID and vice versa, and $\lambda$ is the threshold. Some typically used metrics $S(\mathbf{x})$ include MSP (Hendrycks & Gimpel, 2017), MaxLogit (Hendrycks et al., 2019), Energy (Liu et al., 2020) and KNN (Sun et al., 2022).

**Table 1: Quantitative results of OOD detection performance for ImageNet-1k as ID.** We conduct three pre-training paradigms (ImageNet Pretrained, CLIP, and DINOv2) for comparison. We use FPR95 and AUROC as evaluation metrics. We also report ID classification accuracy.

| | | OOD Datasets | | | | | | | | | | ID ACC |
| | | iNATURALIST | | PLACES | | SUN | | TEXTURES | | Average | | |
| | | FPR95↓ | AUROC↑ | FPR95↓ | AUROC↑ | FPR95↓ | AUROC↑ | FPR95↓ | AUROC↑ | FPR95↓ | AUROC↑ | |
|---|---|---|---|---|---|---|---|---|---|---|---|---|
| ImageNet Pretrained Finetune (SoTA) | Energy (Liu et al., 2020) | 55.72 | 89.95 | 59.26 | 85.89 | 64.92 | 82.86 | 53.72 | 85.99 | 58.41 | 86.17 | 75.08 |
| | MSP (Hendrycks & Gimpel, 2017) | 54.99 | 87.74 | 70.83 | 80.86 | 73.99 | 79.76 | 68.00 | 79.61 | 66.95 | 81.99 | 75.08 |
| | MaxLogit (Hendrycks et al., 2019) | 54.05 | 87.43 | 72.98 | 78.03 | 73.37 | 78.03 | 68.85 | 79.06 | 67.31 | 80.64 | 75.08 |
| | KNN (Sun et al., 2022) | 7.30 | 98.46 | 48.40 | 88.24 | 56.46 | 88.14 | 39.91 | 89.23 | 38.02 | 91.01 | 75.08 |
| | MOS (Huang & Li, 2021) | 9.54 | 98.23 | 43.62 | 91.26 | 48.15 | 90.42 | 57.12 | 83.16 | 39.60 | 90.76 | 75.20 |
| CLIP-Based Methods | Energy (Liu et al., 2020) | 65.00 | 87.17 | 57.40 | 87.32 | 46.43 | 91.17 | 57.40 | 87.32 | 56.55 | 88.24 | 79.39 |
| | MSP (Hendrycks & Gimpel, 2017) | 40.89 | 88.63 | 65.81 | 81.24 | 67.90 | 80.14 | 64.96 | 78.16 | 59.89 | 82.04 | 79.39 |
| | MaxLogit (Hendrycks et al., 2019) | 60.86 | 88.03 | 55.5 | 87.44 | 44.81 | 91.16 | 52.25 | 86.04 | 53.35 | 88.16 | 79.39 |
| | MCM (Ming et al., 2022) | 30.91 | 94.61 | 37.59 | 92.57 | 44.69 | 89.77 | 57.77 | 86.11 | 42.74 | 90.77 | 67.01 |
| | CLIPN (Wang et al., 2023) | 23.94 | 95.27 | 26.17 | 93.93 | 33.45 | 92.28 | 40.83 | 90.93 | 31.10 | 93.10 | 68.53 |
| | LSN (Nie et al., 2024) | 21.56 | 95.83 | 34.48 | 91.25 | 26.32 | 94.35 | 38.54 | 90.42 | 30.22 | 92.96 | 71.89 |
| DINOv2-Based Methods | Energy (Liu et al., 2020) | 13.23 | 96.86 | 66.63 | 83.32 | 61.57 | 84.76 | 66.43 | 82.36 | 51.96 | 86.82 | 81.70 |
| | MSP (Hendrycks & Gimpel, 2017) | 9.05 | 98.15 | 52.58 | 86.34 | 49.45 | 87.35 | 52.32 | 85.82 | 40.85 | 89.41 | 81.70 |
| | MaxLogit (Hendrycks et al., 2019) | 8.21 | 98.22 | 53.93 | 85.80 | 50.48 | 87.00 | 54.32 | 85.25 | 41.73 | 89.06 | 81.70 |
| | KNN (Sun et al., 2022) | 3.01 | 98.26 | 42.78 | 88.89 | 35.96 | 91.51 | 35.30 | 91.05 | 29.27 | 92.67 | 81.70 |
| | Naive finetuning | 5.67 | 97.65 | 43.25 | 88.21 | 36.42 | 90.21 | 28.04 | 92.66 | 28.34 | 92.18 | 85.96 |

## 2.2 IS OOD DETECTION STILL A PROBLEM IN THE ERA OF FOUNDATION MODELS?

Most previous studies have primarily focused on fine-tuning ImageNet-pretrained model (Russakovsky et al., 2015) on ID data for OOD detection. However, ImageNet pre-training has been shown to be relatively outdated in the contemporary development of vision foundation models. In particular, vision foundation models (e.g., DINOv2), which are trained on vast amounts of data, have demonstrated their ability to generate discriminative and generalizable features. This development has inspired us to reexamine this issue within the context of large foundation models. In this section, we aim to investigate and analyze the potential of pre-trained vision foundation models as effective OOD detectors without fine-tuning.

**Experimental Setup.** We perform our evaluation on a challenging OOD detection benchmark that utilizes ImageNet-1K as ID data and selects samples from iNATURALIST, SUN, PLACES, and TEXTURES as OOD samples. We choose two representative vision foundation models, namely CLIP (Radford et al., 2021) and DINOv2 (Maxime et al., 2023). Without any model tuning, we directly use the features extracted from these models for OOD detection evaluation to assess whether they are already sufficiently capable of OOD detection. To emphasize the significance of our findings, we also compare them with state-of-the-art methods that involve fine-tuning an ImageNet pre-trained model on the ID dataset.

**Result Analysis.** As shown in Tab. 1, we compare the OOD detection performance of foundation models without fine-tuning and SOTA ImageNet pre-trained models with fine-tuning, We observe:

**(1)** When using traditional metric scores (*i.e.*, Energy, MSP, and Maxlogit), CLIP does not exhibit superior performance. However, after implementing negative prompts, CLIP-based methods (Wang et al., 2023; Nie et al., 2024) (*e.g.*, LSN (Nie et al., 2024)) outperformed the method (*e.g.*, MOS (Huang & Li, 2021)) by over 9%. Specifically, negative prompts learn the concept of "not this category", opposite to the normal positive prompt "a photo of this category". Since CLIP is trained by a vast amount of categories, it can summarize the concept of "not this category" in its feature space through negative prompts, which helps to establish a more discriminative decision boundary. Please refer to the papers (Wang et al., 2023; Nie et al., 2024) for more details. These results indicate that the raw feature space exhibits good OOD performance since it outperforms the finetuning-based method

without any tuning in the vision encoder. We also visualize the feature space of foundation models for validating the experimental results. In the first column, we randomly select 15 categories from different superclasses in ImageNet-1k. As shown in Fig. 1a and Fig. 1d, both models show compact feature space, *which indicates that foundation models can provide good feature representations for the OOD detection task.*

**(2)** *CLIP is not skilled at distinguishing fine-grained OOD samples while DINOv2 do perform significantly better.* For example, CLIPN (Wang et al., 2023) only achieves 21.56% FPR95 when using iNaturalist18 serves as the OOD samples, while KNN (ImageNet pre-trained model + Finetune) reaches 7.30%. iNaturalist18 provides many hard-to-distinguish fine-grained OOD samples, which are prone to confusion with categories of animals and plants in ImageNet. The reason is that the paradigm of CLIP only provides image-level textual supervision without a supervision signal to retain detailed image information. Therefore, CLIP always fails in some fine-grained tasks, while DINOv2 consistently performs much better. As shown in Fig. 1b and Fig. 1e, where we randomly select 11 fine-grain categories under 3 different super classes, DINOv2 provide more discriminative boundaries, while CLIP can not.

**(3)** *With traditional score metrics (i.e, Energy, MSP, Maxlogit, KNN), DINOv2 can outperform all other methods, surpassing previous methods without any fine-tuning.* DINOv2+KNN shows the best results where the average FPR95 achieves 29.27%. Note that DINOv2+KNN reaches 3.01% FPR95 when using iNaturalist18 as the OOD samples, which indicates that DINOv2 performs very well in fine-grained discrimination. This is because DINOv2 leverages advanced self-supervised learning: iBot (Zhou et al., 2022), which is a Mask Image Modeling (MIM) pretask for facilitating models to capture image details, and DINOv1 (Caron et al., 2021) that enhances the feature discriminativeness by contrastive learning objectives.

**Further Discussion.** In summary, DINOv2, without requiring any fine-tuning, can already function as a high-performing OOD detector, surpassing previous approaches and underscoring *the importance of discriminative and generalizable features for OOD detection*. Besides, as shown in Fig. 1c and Fig. 1f, we also show some failure examples, where the models exhibit particularly poor feature discriminability. It demonstrates that foundation models, even DINOv2, still have room to improve and cannot generalize well across the entire feature space. Moreover, though there is a consensus that fine-tuning on the ID data can improve OOD performance (Vaze et al., 2022; Chen et al., 2020; Hendrycks et al., 2019; Tack et al., 2020), we find that this doesn't hold in the context of foundation models, particularly on in-domain data with large semantic spaces. For instance, when we fine-tune DINOv2 on ImageNet-1K ID data and evaluate the fine-tuned model, the performance declines on three out of the four OOD datasets (see naive finetuning in Tab. 1). The implementation details of this finetuning can be referred to Appx. A.2. These findings motivate us to design specialized fine-tuning methods for vision foundation models to achieve better OOD performance.

## 3 METHOD

This section introduce our proposed methods for finetuning vision foundation models to enhance the OOD detection ability, which includes a Mixture of Feature Expert module in Sec. 3.1 and a Dynamic-$\beta$ Mixup data augmentation strategy in Sec. 3.2.

### 3.1 MIXTURE OF FEATURE EXPERTS

As shown in Fig. 2, we propose Mixture of Feature Experts (MoFE), which divides the complex semantic space into multiple subspaces and each expert specializes in a specific subspace. Each expert can tackle an easier problem instead of conducting OOD detection on a complicated distrubution, which eases the optimization process while maintaining the generalizability of features. Below present the detailed configuration of MoFE.

Given an RGB image $\mathbf{v} \in \mathbb{R}^{H \times W \times 3}$, where $H$ and $W$ are the origin resolution, we reshape the image $\mathbf{x} \in \mathbb{R}^{H \times W \times C}$ into a sequence of flattened 2D patches $\mathbf{x}_p \in \mathbb{R}^{N \times (P^2 \cdot C)}$, $C$ is the number of channels, $(P, P)$ is the resolution of each image patch. Next, we flatten the patches and map to $D$ dimensions with a trainable linear projection $\mathbf{E}$. A learnable embedding is prepended to the sequence of embedded patches ($\mathbf{z}_0^0 = \mathbf{x}_{\text{cls}}^0$) and position embeddings are added to the patch embeddings $E_{pos}$. Then we input these embeddings to multiple transformer blocks. The output is processed by a MoFE

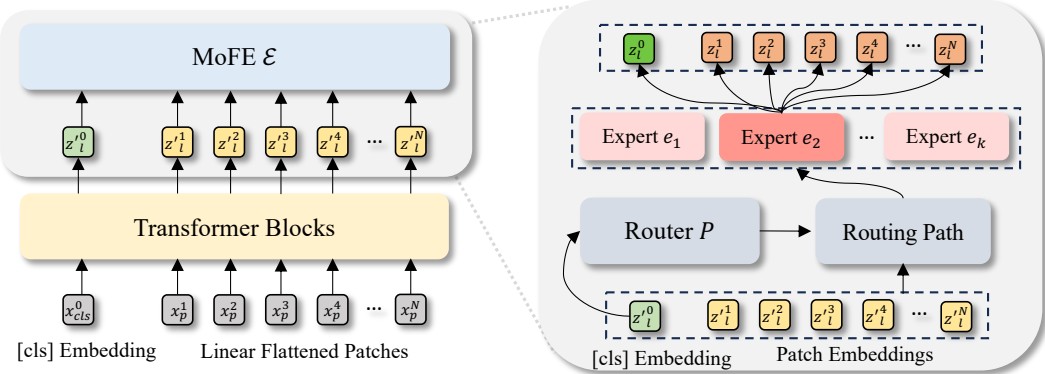

**Figure 2: Illustration of our proposed Mixture of Feature Experts (MoFE).** MoFE decomposes the large semantic space into multiple subspaces and each expert specializes in a specific subspace. Specifically, the image patches and the class token are input to obtain the preliminary patch embeddings and class embedding. A router is employed to determine the expert to further process the embeddings, and the input of the router is the class embedding. Finally, we apply associated experts to refine the class embeddings and the patch embeddings. We use the class embeddings output by MoFE and conduct the OOD detection in the corresponding subspace.

layer to obtain the domain-specific features. This process is expressed as:

$$\mathbf{z}_0 = [\mathbf{x}_{\text{cls}}^0; \mathbf{x}_p^1 \mathbf{E}; \mathbf{x}_p^2 \mathbf{E}; \cdots; \mathbf{x}_p^N \mathbf{E}] + \mathbf{E}_{pos}, \qquad \mathbf{E} \in \mathbb{R}^{(P^2 \cdot C) \times D}, \ \mathbf{E}_{pos} \in \mathbb{R}^{(N+1) \times D} \qquad (2)$$

$$\mathbf{z}'_\ell = \text{Transformer}(\mathbf{z}_{\ell-1}) + \mathbf{z}_{\ell-1}, \qquad \ell = 1 \dots L \qquad (3)$$

$$\mathbf{z}_\ell = \text{MoFE}(\text{LN}(\mathbf{z}'_\ell)) + \mathbf{z}'_\ell, \qquad (4)$$

$$\mathbf{F} = \text{LN}(\mathbf{z}_L). \qquad (5)$$

**MoFE Architecture.** The MoFE layer consists of multiple expert networks, each of which is a transformer block. As an initialization step, we replicate the transformer blocks from the final layer of a foundation model to form an ensemble of experts $\mathcal{E} = [e_1, e_2, \cdots, e_E]$. The router is a linear layer that predicts the probability of each token being assigned to each expert. Routing accuracy is crucial for MoFE. The key question is what should be used to determine the results of feature routing? We explore various approaches, such as reinitializing a routing token, averaging patch embeddings, or utilizing class embeddings. We ultimately find that using the class embedding achieves the best results. Although it is not the embedding output from the last layer of the network, it is sufficiently discriminative. Therefore, we utilize the class embedding $\mathbf{z}'^0_l$ as the input of the router. The router is a linear layer that predicts the probability of each token being assigned to each expert. We formulate as:

$$\mathcal{P}(\mathbf{z}'^0_l)_i = \frac{e^{f(\mathbf{z}'^0_l)_i}}{\sum_j^E e^{f(\mathbf{z}'^0_l)_j}}, \qquad (6)$$

where the router produces weight logits $f(\mathbf{z}'^0_l) = \mathbf{W} \cdot \mathbf{z}'^0_l$, which are normalized by the softmax function. $\mathbf{W} \in \mathbb{R}^{D \times E}$ represents the lightweight training parameters and $E$ represents the number of experts. After determining the experts by using the class embedding, we input all embedding including the patch embeddings and class embedding to the activated experts. Each embedding is processed by the top-$k$ experts with the highest probabilities, and the weighted sum is calculated based on the softmax results of the probabilities:

$$\text{MoFE}(\mathbf{z}'_\ell) = \sum_{i=1}^k \mathcal{P}(\mathbf{z}'^0_l)_i \cdot \mathcal{E}(\mathbf{z}'_\ell)_i, \qquad (7)$$

where $\mathcal{E}$ represents the network of an expert. In our MoFE architecture, we route to only a single expert, thus $k = 1$. We find that the router computation is reduced as we are only routing a token to a single expert and the performance does not increase when using more experts.

**Feature Space Separation.** In MoFE, we aim to have different experts specialize in different subspaces. Therefore, we propose to first separate the whole feature space into multiple subspaces so

that each expert specializes in learning features within its subspace. Specifically, we extract feature representations $\mathbf{z}_l'^0$ for each training image. Then, we calculate the class prototypes by averaging the features of the images from each category. Finally, we perform a K-Means clustering on categorical feature prototypes. Therefore, we explicitly define the route path for each sample. We determine the initial clustering centers based on WordNet semantic information. Each class is associated with a synset in WordNet, from which we can build the taxonomy as a hierarchical tree. We average the class features of all sub-categories from these super categories. Therefore, the class clustering is consistent across multiple runs.

**MoFE Training.** We separate the last $M$ transformer blocks as the MoE layer, where $M = 1$ by default. Then we randomly initialize a router layer and use the class token as the input. We use the pseudo labels generated by the above clustering to supervise the routing:

$$\mathcal{L}_{\text{route}} = -\frac{1}{N} \sum_{n=1}^{N} \sum_{i=1}^{E} y^i \log(\mathcal{P}^i(\mathbf{z}_l'^0)). \tag{8}$$

For each expert, we leverage the categories within the corresponding cluster as the positive samples, and the categories beyond the cluster as the negative ones. Assuming that the category cluster of the $ith$ expert contains $Q_i$ classes, we set the categories beyond the cluster as the $Q_i + 1$ categories. The loss is designed as follows:

$$\mathcal{L}_{\text{expert}} = -\frac{1}{N} \sum_{n=1}^{N} \sum_{i=1}^{E} \sum_{q=1}^{Q_i+1} y^i y^q \log(p_i^q(\mathbf{x})). \tag{9}$$

In order to achieve the sample balance for each cluster, we control the ratio of positive and negative samples as 1:1 during training. Therefore, the overall loss of MoFE is:

$$\mathcal{L}_{\text{MoFE}} = \mathcal{L}_{\text{expert}} + \mathcal{L}_{\text{route}}. \tag{10}$$

### 3.2 Dynamic-$\beta$ Mixup

Data augmentation (*e.g.*, Mixup (Thulasidasan et al., 2019; Zhang et al., 2017)) has been proven to improve generalization during finetuning. Traditional Mixup (Thulasidasan et al., 2019; Zhang et al., 2017) augment samples and transform labels by:

$$\tilde{x} = \lambda x_i + (1 - \lambda)x_j, \quad \tilde{y} = \lambda y_i + (1 - \lambda)y_j, \tag{11}$$

where $\lambda \sim \text{Beta}(\sigma, \sigma)$. $\lambda$ is the interpolation weight for generating new augmented samples. We observe that different categories exhibit varying levels of learning difficulty, since different categories shows different discriminativeness, as shown in Fig. 1. Therefore, we dynamically adjust the Beta distribution according to the feature discriminativeness per category. The reason is that when features of $x_i$ are discriminative enough, a small $\lambda$, which leads to a dissimilar sample, is not necessary for their representation learning. Instead, we should leverage similar samples from a large $\lambda$ for building smooth decision boundaries. On the contrary, when features of a category show poor discriminativeness, we should set a relatively small $\lambda$ to ease the feature learning. We use the accuracy of the validation set to measure the discriminativeness. Therefore, we set $\lambda$ as

$$\lambda \sim \text{Beta}(\sigma, \sigma) \text{ for } \sigma = 1 - w * s, \tag{12}$$

where $w$ is a scaling factor and s denotes the corresponding category's accuracy on the validation set. Because the probability density function of $\text{Beta}(\sigma, \sigma)$ is symmetric about 0.5 and ranges from 0 to 1, we need to ensure that with a larger $s$, the probability of sampling larger values is greater. Therefore, we transform $\lambda$ as,

$$\hat{\lambda} = \begin{cases} \lambda & \lambda \geq 0.5 \\ 1 - \lambda & \lambda < 0.5 \end{cases}. \tag{13}$$

We determine the category difficulty at the beginning of the training and then update it during the training process. In our implementation, $x_i$ is the training sample, and $x_j$ is the instance used to corrupt $x_i$. Therefore, we select the s from categories of $x_i$, and we select samples from different classes. Additionally, we empirically find that using vanilla Mixup (Thulasidasan et al., 2019; Zhang et al., 2017) can cause feature norms to grow during finetuning vision foundation models (*i.e.*,

**Table 2: Quantitative results of OOD detection performance for ImageNet-1k as ID.** We employed our method on two pre-training paradigms (CLIP, and DINOv2). We use FPR95 and AUROC as evaluation metrics. We also report ID classification accuracy.

| | | OOD Datasets | | | | | | | | | | ID ACC |
|---|---|---|---|---|---|---|---|---|---|---|---|---|
| | | iNaturalist | | Places | | Sun | | Textures | | Average | | |
| | | FPR95↓ | AUROC↑ | FPR95↓ | AUROC↑ | FPR95↓ | AUROC↑ | FPR95↓ | AUROC↑ | FPR95↓ | AUROC↑ | |
| CLIP-Based | Energy (Liu et al., 2020) | 65.00 | 87.17 | 57.40 | 87.32 | 46.43 | 91.17 | 57.40 | 87.32 | 56.55 | 88.24 | 79.39 |
| | MSP (Hendrycks & Gimpel, 2017) | 40.89 | 88.63 | 65.81 | 81.24 | 67.90 | 80.14 | 64.96 | 78.16 | 59.89 | 82.04 | 79.39 |
| | MaxLogit (Hendrycks et al., 2019) | 60.86 | 88.03 | 55.5 | 87.44 | 44.81 | 91.16 | 52.25 | 86.04 | 53.35 | 88.16 | 79.39 |
| | MCM (Ming et al., 2022) | 30.91 | 94.61 | 37.59 | 92.57 | 44.69 | 89.77 | 57.77 | 86.11 | 42.74 | 90.77 | 67.01 |
| | CLIPN (Wang et al., 2023) | 23.94 | 95.27 | 26.17 | 93.93 | 33.45 | 92.28 | 40.83 | 90.93 | 31.10 | 93.10 | 68.53 |
| | LSN (Nie et al., 2024) | 21.56 | 95.83 | 34.48 | 91.25 | 26.32 | **94.35** | 38.54 | 90.42 | 30.22 | 92.96 | 71.89 |
| | *Ours* | **17.19** | **97.01** | **24.27** | **94.35** | **22.47** | 94.27 | **35.79** | **91.45** | **24.92** | **94.27** | **73.43** |
| Dinov2-Based | MSP (Hendrycks & Gimpel, 2017) | 25.02 | 94.76 | 57.09 | 83.45 | 53.65 | 85.22 | 48.79 | 85.81 | 48.13 | 87.31 | 86.01 |
| | MaxLogit (Hendrycks et al., 2019) | 22.96 | 94.59 | 59.21 | 78.41 | 54.52 | 81.80 | 48.17 | 84.16 | 46.21 | 84.74 | 86.01 |
| | Energy (Liu et al., 2020) | 28.48 | 93.19 | 65.88 | 74.49 | 61.54 | 78.71 | 53.29 | 81.92 | 52.29 | 82.07 | 86.01 |
| | KNN (Sun et al., 2022) | 5.67 | 97.65 | 43.25 | 88.21 | 36.42 | 90.21 | 28.04 | 92.66 | 28.34 | 92.18 | 86.01 |
| | *Ours* | **2.74** | **98.82** | **24.32** | **93.73** | **17.38** | **95.65** | **18.58** | **95.38** | **17.01** | **95.89** | **86.40** |

DINOv2), leading to performance degradation on the OOD task. In order to restrain the growth of feature norms, we propose to add a regularization term to suppress the increase in feature norm,

$$\mathcal{L}_{\text{Mixup}} = -\frac{1}{N} \sum_{n=1}^{N} \sum_{c=1}^{C} y^c \log(p^c(x)) + Reg(F^0),  \tag{14}$$

where $C$ is the total number of categories, $Reg$ denotes a regularization method, $F^0$ is the final class embeddings output by MoFE. By default, the regularization method has multiple choices, which can be $L_2$ norm or label smoothing.

## 4 EXPERIMENTS

### 4.1 BENCHMARK

**In- and out-distribution Datasets.** To validate the effectiveness of our proposed method, we conduct evaluation on both large-scale and small scale dataset. We use ImageNet-1K (Russakovsky et al., 2015) and ImageNet-100 (Ming et al., 2022) as the ID datasets. Following MOS (Huang & Li, 2021), we consider diverse OOD test datasets, including samples selected from iNaturalist (Van Horn et al., 2018), SUN (Xiao et al., 2010), Places (Zhou et al., 2017), and Textures (Cimpoi et al., 2014).

**Method Comparison.** We conduct method comparison on two pretaining paradigms(*i.e.* CLIP and DINOv2). For each group, we apply some traditional scoring metric (such as MSP (Hendrycks & Gimpel, 2017), MaxLogit (Hendrycks et al., 2019), Energy (Liu et al., 2020), KNN (Sun et al., 2022)). Moreover, we also involve the current CLIP-based state-of-the-art methods, such as MCM (Ming et al., 2022) CLIPN (Wang et al., 2023), and LSN (Nie et al., 2024). We use KNN as the scoring metric when using DINOV2, and follow the scoring metric of CLIPN (Wang et al., 2023) when applying our method to CLIP. More implementation details can be referred to supplementary material.

### 4.2 MAIN RESULTS

**Results on ImageNet-1K.** We compare the proposed approach with the state-of-the-art methods for ImageNet-1K as ID on Tab. 2. These results show: 1) Based on DINOv2, our method reaches the best performance when setting ImageNet-1K as ID. Specifically, our approach reaches 17.01% FPR95 and 95.89% AUROC, averaging the results of all the OOD test sets. our method surpasses the sota method LSN (Nie et al., 2024) by 13.21% in FPR95, and 2.93% in AUROC. 2) When applying our method on CLIP, our method reaches 24.92% and 94.27%, which also outperforms LSN by a large

**Table 3: Quantitative results of OOD detection performance for ImageNet-100 as ID.** We employed our method on two pre-training paradigms (CLIP, and DINOv2). We use FPR95 and AUROC as evaluation metrics. We also report ID classification accuracy.

| | | OOD Datasets | | | | | | | | | | ID ACC |
|---|---|---|---|---|---|---|---|---|---|---|---|---|
| | | iNATURALIST | | PLACES | | SUN | | TEXTURES | | Average | | |
| | | FPR95↓ | AUROC↑ | FPR95↓ | AUROC↑ | FPR95↓ | AUROC↑ | FPR95↓ | AUROC↑ | FPR95↓ | AUROC↑ | |
| CLIP-Based | MSP (Hendrycks & Gimpel, 2017) | 23.55 | 95.92 | 40.46 | 91.23 | 37.02 | 92.45 | 24.40 | 94.90 | 31.43 | 93.63 | 91.93 |
| | MCM (Ming et al., 2022) | 18.13 | 96.77 | 34.52 | 94.36 | 36.45 | 94.54 | 41.22 | 92.25 | 32.58 | 94.48 | 87.88 |
| | CLIPN (Wang et al., 2023) | 4.87 | 98.16 | 13.64 | 96.93 | 13.55 | 97.56 | 15.78 | 93.02 | 11.96 | 96.41 | 91.64 |
| | LSN (Nie et al., 2024) | 4.93 | 98.92 | 12.82 | 97.19 | 8.23 | 97.98 | **8.26** | **98.11** | 8.56 | 98.05 | 92.24 |
| | *Ours* | **3.20** | **99.17** | **10.05** | **97.76** | **7.06** | **98.39** | 9.31 | 97.10 | **7.40** | **98.10** | **92.85** |
| DINOv2-Based | MSP (Hendrycks & Gimpel, 2017) | 5.06 | 98.85 | 26.58 | 94.78 | 27.64 | 95.02 | 26.43 | 94.27 | 21.42 | 95.72 | 94.50 |
| | MaxLogit (Hendrycks et al., 2019) | 5.55 | 98.76 | 29.69 | 94.19 | 32.73 | 94.20 | 29.27 | 93.72 | 24.31 | 95.21 | 94.50 |
| | Energy (Liu et al., 2020) | 18.57 | 96.69 | 54.72 | 88.92 | 62.42 | 87.17 | 57.28 | 88.40 | 48.21 | 90.29 | 94.50 |
| | KNN (Sun et al., 2022) | 2.58 | 99.02 | 18.45 | 95.12 | 15.89 | 96.16 | 16.79 | 96.38 | 13.42 | 96.66 | 94.50 |
| | *Ours* | **2.25** | **99.23** | **12.81** | **96.66** | **8.51** | **97.86** | **8.85** | **97.28** | **8.10** | **97.75** | **96.94** |

**Table 4:** Ablation study of individual components.

| Settings | IN-1K | |
|---|---|---|
| | FPR95↓ | AUROC↑ |
| Baseline | 29.27 | 92.67 |
| + MoFE | 22.59 | 94.01 |
| + D-$\beta$ | 23.85 | 93.72 |
| + MoFE+D-$\beta$ | **17.01** | **95.89** |

**Table 5:** The effect of Cluster Number. We report the performance gain in FPR95 compared to the model without MoFE.

| Num | Gain | Num | Gain |
|---|---|---|---|
| 2 | 4.1 | 7 | 12.26 |
| 3 | 6.3 | 8 | 12.10 |
| 5 | 9.8 | 9 | 12.09 |

margin. These results indicate the effectiveness of the proposed MoFE and the dynamic regularized Mixup. 3) Our approach reaches 2.74% FPR95 on iNaturalist and increases the performance on all the test sets, which indicates that our MoE design retains the discriminativeness of DINOv2 and facilitates feature learning on various feature subspaces.

**Results on ImageNet-100.** We compare the proposed approach with the state-of-the-art methods for ImageNet-100 as ID on Tab. 3. Based on DINOv2, our method reaches 8.10% FPR95 and 97.75% AUROC, surpassing the baseline by 4.40% FPR95, and 0.23% AUROC; This indicates that our proposed approach is also effective in a small-scale ID dataset. On the other hand, when applying our method to CLIP, we achieve 7.40% FPR95 and 98.10% AUROC, outperforming LSN by 1.16% FPR95. The above experimental results validate the effectiveness of our approach, and we can achieve the best performance on both small-scale and large-scale ID datasets.

## 4.3 ANALYSIS

**Contributions of Individual Components.** As shown in Tab. 4, we conduct ablation studies using ImageNet-1K as ID data and report the average performance on the four out-of-distribution datasets mentioned in Sec. 4.1. We follow the same experimental setting in the rest of this section. On ImageNet-1K, MoFE and Dynamic-$\beta$ Mixup contribute 6.68% and 5.41% FPR95, respectively. When combined, the best performance are 17.01% FPR95, and 95.89% AUROC.

**Cluster Number.** We conduct an experiment to validate the impact of cluster number on MoFE performance. We set different numbers of clusters. As shown in Tab. 5, we report the performance gain in FPR95. The results show that as the increasing of cluster number, the performance gradually increases. The performance saturates when the cluster number reaches 7.

**Grouping Srategy.** As shown in Tab. 6, we validate the different strategies for determining the cluster for each cluster. We compare our method with two methods: Taxonomy and self-learning. The results show that using the feature clustering is the most promising approach. The reason might be that the features extracted by pretrained model are already discriminative enough, especially at coarse-grained level. Therefore, the feature similarity can be used to determine the cluster.

**Table 6:** Analysis on Grouping Strategy in MoFE.

| Grouping | FPR95↓ | AUROC↑ |
|---|---|---|
| Baseline | 29.27 | 92.67 |
| Taxonomy | 25.63 | 93.01 |
| Self-Learning | 26.34 | 92.99 |
| Ours | **22.59** | **94.01** |

**Table 7:** Ablation study of Dynamic-$\beta$ Mixup.

| Methods | FPR95↓ | AUROC↑ |
|---|---|---|
| Baseline | 29.27 | 92.67 |
| w/o Reg | 30.43 | 91.65 |
| w/o D-$\beta$ | 24.96 | 93.36 |
| Ours | **23.85** | **93.72** |

**More Analysis on Dynamic-$\beta$ Mixup.** As shown in Tab. 7, we conduct an ablation study on Dynamic-$\beta$ Mixup. When we remove the regularization term, we find that the performance degrades (30.43% FPR95). Moreover, when we dynamic beta distribution is removed, the performance decreases to 24.96% FPR95.

## 5 RELATED WORK

**Out-of-Distribution Detection** The goal of OOD detection is to detect OOD images from the test dataset (containing both ID and OOD images). Designing the score function is the most popular method in OOD detection tasks. The scores are mainly derived from three sources: the probability (Hendrycks & Gimpel, 2017; Hendrycks et al., 2019), the logits (Hendrycks et al., 2019; Liu et al., 2020), and the feature (Lee et al., 2018b; Ndiour et al., 2020). Some studies (Khalid et al., 2022; Wang et al., 2022b; Sehwag et al., 2021) focus on leveraging contrastive learning to enhance the feature representation. Other studies show that synthesizing pseudo samples (Du et al., 2022; Sehwag et al., 2021; Tack et al., 2020;?) as OOD instances is also a promising approach to make the feature space more compact.

**OOD Detection with Foundation Models** There are some existing OOD detection methods (Wang et al., 2023; Esmaeilpour et al., 2022; Ming et al., 2022; Ming & Li, 2024; Nie et al., 2024) leveraging foundation models. Maximum Concept Matching (MCM) (Ming et al., 2022) proposes a simple yet effective zero-shot OOD detection method by aligning visual features with textual concepts. Some other studies (Wang et al., 2023; Nie et al., 2024) explore negative prompts to learn the diversity of negative features, enabling more accurate detection of OOD samples. Although these studies have made great progress by leveraging CLIP to enhance the performance in existing benchmarks, they only explore and fine-tune CLIP. In our studies, we explore different foundation models and explore a better fine-tuning paradigm.

**Mixture of Experts** Mixture of Experts has been studied independently in both computer vision (Riquelme et al., 2021; Lou et al., 2021; Mustafa et al., 2022) and natural language processing (Shazeer et al., 2017; Lepikhin et al., 2020; Fedus et al., 2021; Komatsuzaki et al., 2022). These works are studied in the context of conditional computation, which is to increase the number of model parameters without a proportional increase in computational cost. Currently, some studies (Chen et al., 2024; Krishnamurthy et al., 2023) explore improving expert specialization and leveraging MoE to mitigate data conflict problems, where some data might interfere with each other. In our study, we introduce MoFE to the out-of-distribution task in the context of foundation models and build specialized OOD detectors for different feature subspaces.

## 6 CONCLUSION

This paper studies the OOD detection task within the context of foundation models. Our study shows that vision foundation models (e.g., DINOv2) are effective OOD detectors, suggesting high-quality and generalizable feature space is essential for OOD detection. our study highlights that CLIP's pre-trained feature space is less effective for fine-grained tasks like iNaturalist18, where DINOv2 performs significantly better, which worths further exploration. Second, we find that simply fine-tuning foundation models on ID data will result in performance degradation due to the loss of generalization ability. Thus, we propose MoFE and a Dynamic-$\beta$ Mixup data augmentation to enhance the feature learning during fine-tuning. We conduct extensive experiments and ablation studies to validate the effectiveness of our approach, significantly surpassing baseline methods. We believe enhancing the discriminativeness and generalization ability of learned features is the key to OOD detection. We hope our investigation could inspire more future studies.

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

# A   APPENDIX

## A.1   IMPLEMENTATION DETAILS

We adopt ViT-Base (Dosovitskiy et al., 2021) as the backbone. When using pre-training paradigms of CLIP and DINOv2, we directly initialize ViT from their weights. Besides, when using CLIP, we leverage CLIPN (Wang et al., 2023) as the baseline method and we follow their scoring metric. For DINOv2, we use DINOv2 with standard cross-entropy loss as the baseline method and the scoring metric is KNN (Sun et al., 2022). When using DINOv2, we first conduct linear probing for 3 epoches to ensure its training stability. Our models are trained with AdamW optimizer (Loshchilov & Hutter, 2019) with $\beta_s = \{0.9, 0.95\}$, with an effective batch size of 1024 on 8 NVIDIA 3090 GPUs. The values for weight decay and layer decay are 0.05 and 0.75, The training epochs are set to 40. We set a cosine learning rate schedule and the minimum learning rate is 1e-6.

## A.2   IMPLEMENTATION DETAILS OF NAIVE FINETUNING

The model is trained with cross entropy loss and Adam optimizer with $\beta_s = \{0.9, 0.95\}$, with an effective batch size of 1024 on 8 NVIDIA 3090 GPUs. We use cThe values for weight decay and layer decay are 0.05 and 0.75. The training epochs are set to 40. We set a cosine learning rate schedule, and the minimum learning rate is 1e-6. We first conduct linear probing for 3 epoches to ensure their training stability. During the testing phase, we use KNN as the classifier using features from the penultimate layer.

## A.3   LIMITATION

We summarize the limitations of our research as follows: Although CLIP and DINOv2 are currently the top foundation models, they still have inherent shortcomings. For instance, CLIP only utilizes image-text pairs for contrastive learning between text and images, lacking self-supervised learning on images. This results in its inability to capture fine-grained image details, leading to poor performance on granularity. On the other hand, DINOv2 employs a large number of images for self-supervised learning, yet it still performs poorly on certain categories, indicating potential long-tail distribution issues in its pre-training data. The current benchmarks for OOD (Out-of-Distribution) detection have significant limitations. While they utilize datasets like ImageNet-1K, which cover a wide range of categories, the OOD data itself is relatively limited.

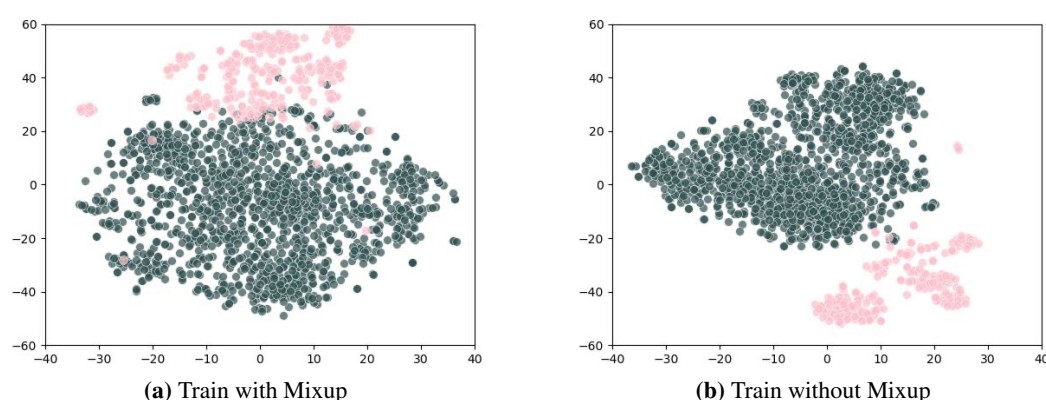

**(a)** Train with Mixup  **(b)** Train without Mixup

**Figure 3: The effect of vanilla mixup on the feature space of DINOv2.** We can observe that vanilla Mixup can blur the decision boundary between ID and OOD.

.

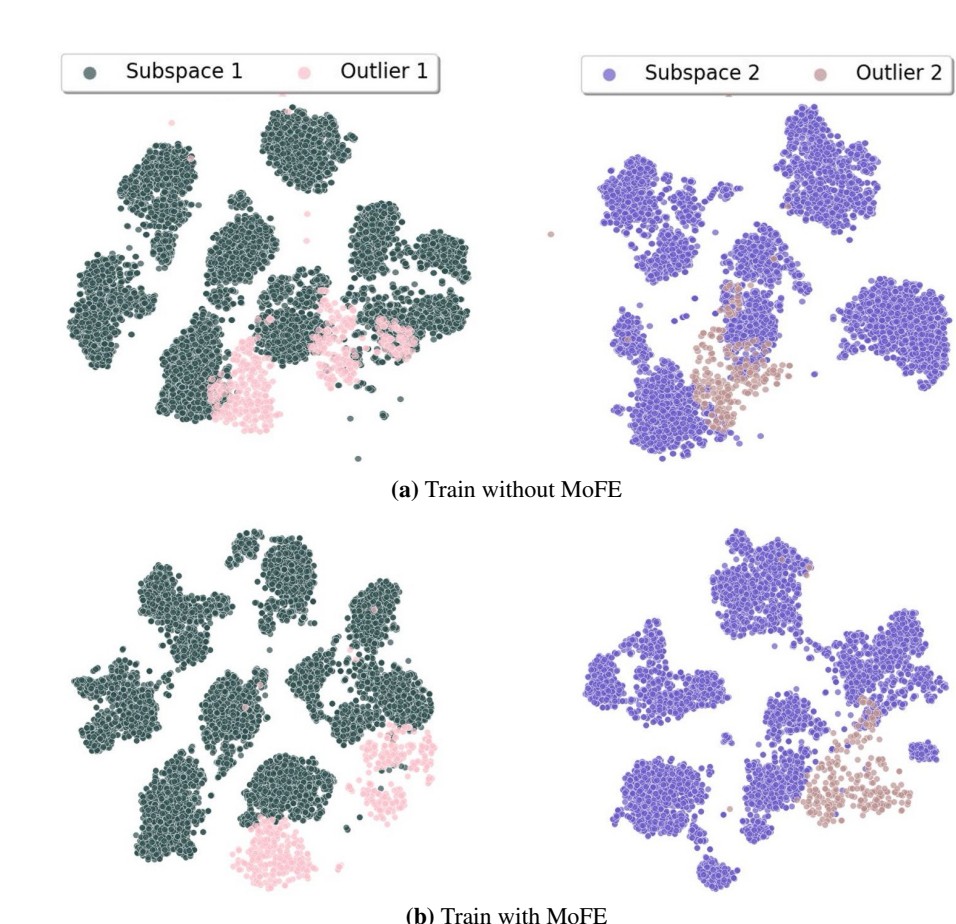

(a) Train without MoFE

(b) Train with MoFE

**Figure 4: Visualization of feature space of MoFE.** It can be observe that, without MoFE, the outlier features are mingled with in-domain data, while MoFE can well separate the in- and out-of-distribution data.

