# OpenReview forum: "Rethinking Out-of-Distribution Detection in Vision Foundation Models"
_ICLR.cc/2025/Conference — ICLR 2025 Conference Withdrawn Submission_

### Official Review · Reviewer_TtBs · 2024-10-27

**Soundness:** 2
**Presentation:** 2
**Contribution:** 2
**Rating:** 3
**Confidence:** 5

**Summary:**

This paper conduct a comprehensive study on pre-trained foundation models for OOD detection. The authors argues that foundation models like CLIP and DINOv2, despite their strong feature learning capabilities, have been underexplored in OOD detection tasks. Through extensive experiments, they demonstrate that a pre-trained DINOv2 model, even without fine-tuning, can outperform previous state-of-the-art OOD detection models, especially when using a simple scoring metric like KNN. The paper also introduces a novel "Mixture of Feature Experts" (MoFE) module, which partitions the feature space into subspaces to facilitate more effective decision boundary learning during fine-tuning on in-domain (ID) data. To further enhance feature learning, the authors propose a "Dynamic-β Mixup" strategy that adaptively adjusts interpolation weights based on the varying difficulty levels of different categories.

**Strengths:**

- The introduction of MoFE and Dynamic-β Mixup adds a novel perspective on improving OOD detection by optimizing the feature space and handling complex data distributions more effectively.
- The paper evaluates its methods across multiple datasets and shows significant improvements in OOD detection metrics (like FPR95 and AUROC) over previous models.
- By comparing DINOv2 and CLIP, the paper provides insights into the limitations of existing foundation models and highlights DINOv2’s superior performance, particularly on fine-grained OOD tasks.

**Weaknesses:**

- The MoFE method proposed in this paper shares similarities with HVCM[1]. Can the authors provide a more detailed comparison, highlighting the key differences and advantages of MoFE over HVCM?
- While the authors contribute a comprehensive analysis of foundation models in OOD detection, the referenced methods seem incomplete. Notably, LoCoOp[2] and NegLabel[3], two high-performing methods based on CLIP, are not included. Additionally, ReAct[4] and GradNorm[5], commonly used in traditional methods, are also absent. This raises questions about the comprehensiveness and reliability of the authors' foundation model analysis.
- The introduction mentions CLIP's limitations in classifying fine-grained OOD samples, exemplified by iNaturalist. However, NegLabel achieves impressive performance on iNaturalist using CLIP with negative labels. Does it prove that the author's analysis maybe biased or incomplete?
- The introduction of MoFE and Dynamic-β Mixup increases model complexity, which may not be practical for all use cases where computational resources are limited.
- While the paper acknowledges CLIP's limitations, it primarily focuses on DINOv2's performance. More in-depth analysis or improvements specifically tailored to CLIP could strengthen the study's comprehensiveness.
-The dataset "naturalist18" is mentioned multiple times throughout the article, but the experimental section only describes "inaturalist." It's essential to clarify whether these two datasets are the same or different.



[1] Li J, Zhou X, Guo P, et al. Hierarchical Visual Categories Modeling: A Joint Representation Learning and Density Estimation Framework for Out-of-Distribution Detection[C]//Proceedings of the IEEE/CVF International Conference on Computer Vision. 2023: 23425-23435.

[2] Miyai A, Yu Q, Irie G, et al. Locoop: Few-shot out-of-distribution detection via prompt learning[J]. Advances in Neural Information Processing Systems, 2024, 36.

[3] Jiang X, Liu F, Fang Z, et al. Negative label guided ood detection with pretrained vision-language models[J]. arXiv preprint arXiv:2403.20078, 2024.

[4] Sun Y, Guo C, Li Y. React: Out-of-distribution detection with rectified activations[J]. Advances in Neural Information Processing Systems, 2021, 34: 144-157.

[5] Huang R, Geng A, Li Y. On the importance of gradients for detecting distributional shifts in the wild[J]. Advances in Neural Information Processing Systems, 2021, 34: 677-689.

**Questions:**

See Weaknesses.

---

### Official Review · Reviewer_byQ4 · 2024-10-27

**Soundness:** 2
**Presentation:** 2
**Contribution:** 2
**Rating:** 5
**Confidence:** 4

**Summary:**

This work explores the effect of Pre-Trained Vision Models on OOD detection, and proposes two methods to provide OOD detection capabilities. The authors first analyzed the differences between CLIP and DINO V2 on OOD detection performance and gave some conclusions. The authors then proposed Mixture of Feature Experts (MOFE) and Dynamic-β Mixup to enhance the ability of Pre-Trained Vision Models. The experiment illustrates the effectiveness of the method.

**Strengths:**

1. The author's analysis of the impact of pre-training visual models on OOD detection performance is conducive to the development of follow-up work in the field.

2. This paper highlights the excellent performance of Dino V2 in OOD detection.

3. The proposed method is to plug and play.

**Weaknesses:**

1. Although this work analyzes the difference between CLIP and DINO v2, it does not describe it in detail or highlight what problems with DINO v2 have any problems or challenges to OOD, which limits the contribution of this work.

2. The method is increment. MOFE is the improvement of MOE and does not explain the relationship or difference between MOE. Dynamic-β mixup is a variant after Mixup introduces dynamic weights. The most important thing is that these two methods are universal, and can insert any method, which is not strong with the previous analysis.

3. DINO v2's training dataset may be similar to the OOD detection dataset. For example, Dino V2 has been pre-training on ImageNet, which may be why DINO v2 has a good effect. But this paper did not mention it at all. In addition, the training data of the CLIP is the image-text pair, which itself is noisy, and it is recognized in terms of fine-grained tasks.

4. So based on the above analysis, the author wants to express, is the selection of visual foundation models is more important than the OOD method itself?

5. What is the MOE mentioned by LINE 332 and 461? MOFE?

**Questions:**

See Weaknesses.

---

### Official Review · Reviewer_vTsA · 2024-11-02

**Soundness:** 3
**Presentation:** 4
**Contribution:** 3
**Rating:** 5
**Confidence:** 4

**Summary:**

The paper focuses on out-of-distribution detection using pre-trained vision foundation models, such as DINOv2 and CLIP. To address the challenges of fitting complex data distributions, the authors propose a Mixture of Feature Experts (MoFE) module, which partitions features into a series of subspaces with simpler distributions. Additionally, a dynamic mixup strategy based on category discriminability is introduced to further enhance feature learning. The improvements in both in-domain classification and OOD detection are well supported by comparative experiments and ablation studies.

**Strengths:**

1. Significant performance gains are achieved on popular OOD benchmark datasets.
2. Using Mixture of Experts (MoE) for feature enhancement and partitioning is interesting and novel.
3. The paper is well-written and easy to follow.

**Weaknesses:**

1. If I understand correctly, the proposed framework essentially functions as a two-stage classification process, where the first stage classifies among the superclasses (path routing) and the second stage classifies fine-grained subclasses within the identified superclass. The proposed framework shares some similarities with [1] in terms of hierarchical classification. Could you clarify the differences from [1]?
2. Breaking the entire in-distribution (ID) features into multiple subspaces is essential for reducing the difficulty of fitting complex data distributions; however, performance saturates with an increased number of subspaces, resulting in no further performance gains. Does the optimal number of clusters depend on the distribution of the ID dataset? More ablation studies on other ID datasets (e.g., ImageNet-100) are needed, along with necessary explanations regarding how and why performance changes as the number of clusters increases.

[1] Linderman, Randolph, Jingyang Zhang, Nathan Inkawhich, Hai Li, and Yiran Chen. "Fine-grain inference on out-of-distribution data with hierarchical classification." In Conference on Lifelong Learning Agents, pp. 162-183, 2023.

**Questions:**

1. Is it essential to initialize the class clustering based on the WordNet taxonomy, which requires the names of ID classes? What would happen if K-means were directly performed on the categorical feature prototypes instead?
2. What is the accuracy of the router for ID data? What would occur if the router incorrectly assigns an expert for ID data?

---

### Official Review · Reviewer_KNEx · 2024-11-04

**Soundness:** 3
**Presentation:** 3
**Contribution:** 3
**Rating:** 6
**Confidence:** 4

**Summary:**

This paper explores the task of out-of-distribution (OOD) detection in the context of foundational models. They indicate that DINOv2 are effective OOD detectors, highlighting the importance of having a high-quality and generalizable feature space for this task. Additionally, They observe that merely fine-tuning foundation models on in-distribution (ID) data can lead to a decline in performance due to a reduction in generalization ability. To address this, they introduce the Mixture of Feature Experts (MoFE) approach and a Dynamic-β Mixup data augmentation strategy to enhance feature learning during fine-tuning.

**Strengths:**

1. The paper is well-written.
2. This work explores the OOD capabilities of vision foundation models and proposes some strategies based on the findings.
3. The paper has been validated on different datasets, all proving the effectiveness of the proposed method

**Weaknesses:**

1. The paper also lacks some deeper mine, such as why there are differences between the CLIP and DINO pre-trained models on OOD tasks. For the OOD detection task, what kind of architecture or training approach leads to the most effective pre-trained models?
2. The author constructed the expert network using transformer blocks, which is equivalent to adding an additional layer to the original transformer architecture and increasing the network's depth. Could you explore different configurations for the expert networks to further demonstrate the effectiveness of this framework?
3. The author could provide a more convincing explanation for why foundation models generally outperform traditional models in OOD tasks. While Table 1 presents specific metrics, a deeper analysis is needed to contextualize these results and connect them to the subsequent improvements.
4. Exploring other foundational model architectures as part of the study would be beneficial.

**Questions:**

1. For the OOD detection task, what types of architectures or training approaches contribute to the development of the most effective pre-trained models?
2.  Could you investigate different configurations for the expert networks to better demonstrate the effectiveness of this framework?
3.  While Table 1 presents specific metrics, a more in-depth analysis is required to contextualize these results and connect them to the subsequent improvements.
4. It would be beneficial to explore other foundational model architectures as part of this study.

---

### Author Response · Authors · 2024-11-15

We sincerely appreciate the reviewers for dedicating their time and effort to reviewing our work and for recognizing the potential contributions we may have made. The insightful feedback, constructive comments, and suggestions provided by the reviewers have significantly enhanced the quality and clarity of our work. We will incorporate these valuable suggestions into the revised version to further improve the content.

---

### Note · Authors · 2024-11-15

I have read and agree with the venue's withdrawal policy on behalf of myself and my co-authors.